# Accessible Metaverse: A Theoretical Framework for Accessibility and Inclusion in the Metaverse

Achraf Othman [1,*], Khansa Chemnad [1], Aboul Ella Hassanien [2], Ahmed Tlili [3], Christina Yan Zhang [4], Dena Al-Thani [5], Fahriye Altınay [6], Hajer Chalghoumi [7], Hend S. Al-Khalifa [8], Maisa Obeid [9], Mohamed Jemni [10], Tawfik Al-Hadhrami [11] and Zehra Altınay [6]

1 Mada Qatar Assistive Technology Center, Doha P.O. Box 24230, Qatar; kchemnad@mada.org.qa
2 Faculty of Computers and AI, Cairo University, Cairo 12613, Egypt; aboitcairo@cu.edu.eg
3 Smart Learning Institute, Beijing Normal University, Beijing 100875, China; ahmedtlili@ieee.org
4 The Metaverse Institute, 20-22 Wenlock Road, London N1 7GU, UK; christina@metaverse-institute.org
5 College of Science and Engineering, Hamad Bin Khalifa University, Doha P.O. Box 34110, Qatar; dalthani@hbku.edu.qa
6 Social Research and Development Center, Near East University, Nicosia 99138, Cyprus; fahriye.altinay@neu.edu.tr (F.A.); zehra.altinaygazi@neu.edu.tr (Z.A.)
7 Kids Brain Health Network, 8900 Nelson Way, Burnaby, BC V5A 1S6, Canada; hajer.chalghoumi@hotmail.com
8 Information Technology Department, King Saud University, P.O. Box 145111, Riyadh 4545, Saudi Arabia; hend.alkhalifa@gmail.com
9 Expo City, Dubai, United Arab Emirates; maisa.obeid1@gmail.com
10 Arab League Educational, Cultural and Scientific Organization, Tunis 1003, Tunisia; mohamed.jemni@alecso.org.tn
11 Computer Science Department, School of Science and Technology, Nottingham Trent University, Clifton Lane, Nottingham NG11 8NS, UK; tawfik.al-hadhrami@ntu.ac.uk
* Correspondence: aothman@mada.org.qa

**Abstract:** The following article investigates the Metaverse and its potential to bolster digital accessibility for persons with disabilities. Through qualitative analysis, we examine responses from eleven experts in digital accessibility, Metaverse development, disability advocacy, and policy formulation. This exploration uncovers key insights into the Metaverse's current state, its inherent principles, and the challenges and opportunities it presents in terms of accessibility. The findings reveal a mixed state of inclusivity within the Metaverse, highlighting significant advancements along with notable gaps, especially in integrating assistive technologies and ensuring interoperability across different virtual environments. This study emphasizes the Metaverse's potential to revolutionize experiences for individuals with disabilities, provided that accessibility is embedded in its foundational design. Ethical and legal considerations, such as privacy, non-discrimination, and evolving legal frameworks, are identified as critical factors that shape an inclusive Metaverse. We propose a comprehensive framework that emphasizes technological adaptation and innovation, user-centric design, universal access, social and economic considerations, and global standards. This framework aims to guide future research and policy interventions to foster an inclusive digital environment in the Metaverse. This paper contributes to the emerging discourse on the Metaverse and digital accessibility, offering a nuanced understanding of its complexities and a roadmap for future exploration and development. This underscores the necessity of a multi-faceted approach that incorporates technological innovation, user-centered design, ethical considerations, legal compliance, and continuous research to create an inclusive and accessible Metaverse.

**Keywords:** Metaverse; accessibility; framework; research agenda

## 1. Introduction

The concept of the Metaverse has garnered considerable attention recently, particularly owing to substantial advancements in virtual and augmented reality technologies. The

Metaverse is an integrative ecosystem of virtual worlds offering immersive experiences to users that modify pre-existing experiences and create new value from economic, environmental, social, and cultural perspectives. It is defined as a continuous and interconnected blend of physical and virtual realms facilitated by technologies such as virtual reality and augmented reality, where users engage in multisensory interactions within persistent and shared immersive environments [1]. In this interconnected digital realm, users can engage in real-time interactions with each other and with computer-generated environments, resulting in the blurring of boundaries between the physical and digital worlds [2]. Often described as a fully immersive and interactive space, the Metaverse offers users the opportunity to explore, create, and participate in diverse activities [3]. Neal Stephenson first coined the term Metaverse in his 1992 science fiction novel "Snow Crash" [4]. This concept has transcended the realm of fiction. It is becoming a reality due to rapid technological advancements and the increasing popularity of virtual reality (VR) and augmented reality (AR) experiences. With the Metaverse, individuals can interact and engage with others from around the world, thereby eradicating the barriers of physical distance [5]. This virtual reality space presents infinite possibilities for communication and collaboration, facilitating seamless teamwork and innovation across various industries. The Metaverse in education offers potential applications for blended learning, language learning, competency-based education, and inclusive education but faces challenges and requires further research [6]. Furthermore, the Metaverse could transform the entertainment industry by providing a new level of immersive experiences and redefining how we consume media. Individuals with autism or intellectual disabilities can use it to enhance their communication and social skills [7]. The ongoing development of the Metaverse is expected to significantly transform our daily lives, professional pursuits, and recreational activities, thereby heralding a new era of interconnectedness and virtual exploration. The Metaverse, as envisioned by some experts, is a fully simulated virtual world, while others see it as an augmented reality overlay on the real world [8]. The virtual Metaverse is anticipated to be widely used for gaming, socializing, shopping, entertainment, business, and education. The augmented Metaverse, on the other hand, is regarded as a substitute for mobile phones and a gateway to digital content [9]. The potential application of blockchain technology in decentralizing and resolving control issues in the Metaverse is currently under discussion [10]. However, concerns have also been raised regarding the cost of access, the potential for misinformation, the digital divide, object and identity persistence, and the possible negative impact on real-world relationships and obligations [11,12].

The concept of the Metaverse offers tremendous potential for fostering a more inclusive and accessible digital realm. Although some initiatives have provided recommendations for enhancing the accessibility of virtual and augmented reality content [13,14], complete accessibility solutions seem to be relatively absent. Ensuring equal opportunities for individuals with disabilities in virtual environments entails addressing a variety of obstacles. One significant challenge is the lack of accessibility across numerous virtual reality (VR) platforms and applications [15]. For instance, individuals with visual disabilities may encounter difficulties navigating virtual environments that rely heavily on visual cues, underscoring the need for features such as audio descriptions or haptic feedback. Without such provisions, their ability to fully engage in digital exploration of the Metaverse is impeded. Similarly, those with mobility disabilities may face challenges in VR platforms requiring intricate physical movements or not supporting assistive devices. The use of extended reality (XR) technologies can result in sensory overload for neurodivergent individuals, and such individuals may face difficulties in accessing XR technologies due to the lack of integration of accessibility requirements in current software development practices [16]. Addressing challenges related to the Metaverse for educational purposes involves considering potential accessibility and integration difficulties for individuals with learning disabilities [6]. As the Metaverse continues to develop, it is essential to prioritize and implement inclusive design practices to accommodate the diverse needs of all users. This emphasis on inclusivity can transform the Metaverse into a space in which everyone can actively participate in

virtual experiences, regardless of their abilities. Despite extensive research on VR and augmented reality (AR) for disability [17], translating these advancements into widely available market-ready products appears to be relatively infrequent.

In recent times, the growth of virtual reality platforms has gained considerable momentum, attributed to the efforts of companies such as Oculus and HTC Vive, which have expanded the scope of immersive experiences [18]. Despite the potential of these technologies for entertainment, communication, and education, numerous barriers still hinder individuals with disabilities from fully experiencing virtual environments [15]. This issue highlights the importance of inclusivity and accessibility in the concept of the Metaverse, a shared virtual space. Understanding and addressing these barriers is crucial to ensure that the Metaverse is accessible to all individuals, regardless of their abilities or disabilities. Article 9 of the Convention on the Rights of Persons with Disabilities, adopted by the United Nations, stipulates that all member states must ensure the accessibility of information and communication technologies (ICT) as a fundamental principle [19]. This research is critical in creating equitable opportunities for individuals with disabilities to participate in and benefit from the Metaverse. It is essential to develop technologies and design interfaces that accommodate various disabilities, such as visual, hearing, and physical disabilities. Moreover, collaboration with disability advocacy groups and individuals with disabilities themselves is vital to ensuring that their perspectives and needs are incorporated into the development and design process. Prioritizing accessibility will enable the creation of a Metaverse that fosters inclusivity, diversity, and equal participation for all users [20]. In the same vein, [21] conducted a systematic and bibliometric analysis of the Metaverse and highlighted that less attention is paid to investigating the use of the Metaverse for people with disabilities, calling for more research in this regard.

Considering the background presented, this study primarily aims to investigate the current state of accessibility and inclusivity in the Metaverse through expert opinions and analysis of existing platforms and technologies. While acknowledging the importance of diverse user experiences, our study, at this stage, does not include direct data from people with disabilities regarding their experiences in the Metaverse. Recognizing this as a significant limitation, we intend to address this gap in future research phases, where we will specifically focus on collecting and analyzing firsthand experiences from individuals with disabilities to further our understanding and enhance the inclusivity of the Metaverse. Furthermore, we will explore the role of developers, policymakers, and users in shaping the future of the Metaverse to ensure its accessibility to all. Our research aspires to contribute to the ongoing discourse and initiatives towards constructing a Metaverse that is truly inclusive and accessible to everyone. Through thoroughly examining existing technologies and considering user experiences, we can identify the barriers and constraints that prevent specific individuals from fully participating in the Metaverse. We are convinced that by involving developers, policymakers, and users in this discussion, we can collaboratively tackle these challenges and develop solutions prioritizing inclusivity. Ultimately, our objective is to create a Metaverse that embraces diversity and affords equal opportunities for all individuals to interact, connect, and thrive. Previous research has explored various aspects of the Metaverse, including its technological infrastructure, user experience, and social implications [22,23]. Despite the growing interest in the Metaverse, there is a lack of research on accessibility and inclusion in this virtual world.

This study aims to develop a theoretical framework for accessibility and inclusion in the Metaverse. We propose that a comprehensive understanding of accessibility and inclusion in the Metaverse can be achieved by developing a theoretical framework that incorporates the perspectives of users, designers, and policymakers.

## 2. Background

The concept of the Metaverse has captured the imagination of researchers and technologists alike, offering a glimpse into a future where virtual and physical realities converge. It has gained significant attention in recent years, with much speculation about its potential to

transform how we interact with technology and each other. Despite its growing popularity, the Metaverse is still in its nascent stages of development, and there is no clear consensus on its eventual form. According to Ref. [24], the Metaverse is a proposed future virtual universe that is characterized by the integration of various digital platforms and the provision of immersive experiences through multidimensional interactions and is envisioned as a space that includes augmented, extended, and virtual realities, and is intended to offer users a rich and engaging experience. The current vision for the Metaverse is deeply rooted in the values and preferences of Generation Z, a demographic group known for its fluid online and offline identities. It is reinforced by deep learning-based recognition models and virtual currency [25].

The creation of an immersive and interactive Metaverse depends on the development of virtual reality (VR), augmented reality (AR), artificial intelligence (AI), blockchain, and other technologies [26]. These technologies enable virtual reality experiences, the ownership of digital assets, and decentralized governance. Their implementation is of utmost importance in achieving the desired outcome of a Metaverse. Corporations such as Meta (formerly Facebook), Microsoft, and Epic Games are allocating significant resources toward developing and advancing Metaverse technologies and platforms [27]. This increased investment expedites the progress of these technologies and draws additional interest and attention to the Metaverse. The Metaverse provides a decentralized virtual identity, immersive experiences, interactive functions, a mature economic system, the ability for arbitrary creation, continuous self-evolution, and upgrading. It holds significant potential for growth in social interactions, office environments, education, and gaming [28]. The study of the Metaverse has made notable progress in four key areas: technological development, practical applications, marketing and consumer behavior, and environmental sustainability [29]. These findings hold valuable implications for businesses seeking to capitalize on the potential of the Metaverse. The Metaverse has been studied for its potential in propaganda, solving environmental problems, and achieving sustainable development goals [30]. It has also been studied in education, focusing on topics like roadmaps, dimensions, training, current initiatives, and data [31]. The Metaverse can be used in education for people with disabilities by integrating various information technologies into smart education ecosystems, forming new education modes like virtuality-reality symbiosis, trans-spatial fusion, and collaborative inquiry [22], while also enabling technologies like extended reality and the internet of everything to impact educational services [32]. The Metaverse is a social ecosystem that interconnects the physical and virtual realms, thereby eliciting inquiries about identity, privacy, and security [33]. In the context of the Metaverse, it is of utmost importance to guarantee a secure and safe environment as users participate in social and economic activities within this virtual space. Furthermore, the need for high synchronization and low latency arises as a critical factor in enhancing the user experience and fostering a sense of immersion.

Accessibility and inclusion are critical factors that cannot be disregarded in the realm of the Metaverse [34]. The growth and development of digital environments necessitate the guarantee of equal opportunities for all individuals with different abilities and backgrounds to participate and prosper within them [35]. It is crucial to foster an inclusive environment in the Metaverse, where barriers are eliminated, and every individual has the opportunity to voice their opinions and establish their presence [36]. The Metaverse presents an opportunity to ensure equitable access for all humans, including people with disabilities. Despite the uncertainty of current options available for these individuals in the Metaverse, our research explores potential augmentations and inclusions that can be implemented to ensure their participation in virtual space. It is imperative that people with disabilities are not excluded from the Metaverse. In examining digital accessibility for individuals with disabilities, the prevalent challenges identified were not only technical issues, security and privacy concerns, and operational hurdles but also a notable absence of case studies or practical examples addressing the gap in creating an accessible Metaverse. This omission underscores the need for real-world applications and solutions to better understand and

tackle these barriers. Evidence in the literature indicates a prevalent focus on AI-driven digital accessibility for visual disabilities, which highlights a significant gap in addressing other types of disabilities [37]. Individuals with disabilities often experience limited access to information and communication technology (ICT) and internet connectivity, which can be attributed to various factors such as financial constraints, physical impediments, low levels of digital literacy, and inadequate support [38]. This underscores the pressing need for a realignment of efforts towards a more comprehensive examination of disabilities, urging researchers to broaden their scope and enhance data collection efforts involving individuals with other disabilities, mainly physical/mobility and cognitive. The identified shortcomings in existing systems with respect to adherence to accessibility standards serve to emphasize the pressing need for a fundamental shift in the design of solutions that prioritize the needs of individuals with disabilities. The current web accessibility standards for Metaverse platforms are insufficient in providing digital accessibility for individuals with disabilities [39]. This is due to the absence of specific guidelines and best practices for Metaverse platforms [40] and the limited knowledge of how to modify existing accessibility features to suit virtual environments. Many countries do not have appropriate laws and regulations to ensure the safety and well-being of all participants [41]. To address these challenges, it is essential to thoroughly understand the specific needs and prerequisites of individuals with disabilities to develop effective accessibility solutions for the Metaverse.

The absence of awareness and understanding of accessibility requirements among Metaverse developers can lead to obstacles and limitations within the virtual world [36]. A lack of comprehensive understanding of accessibility can result in unintentional exclusion of certain groups of people from fully participating in the Metaverse experience [34]. This can lead to feelings of exclusion and frustration, particularly for individuals with disabilities who may encounter difficulties navigating or interacting with the virtual environment. Furthermore, the lack of awareness can perpetuate societal biases and stereotypes, further marginalizing already underrepresented communities [42]. Thus, it is imperative that education and training on accessibility become an integral component of the development process so that Metaverse platforms are designed with inclusivity in mind. As technology continues to advance and the virtual world becomes increasingly integrated into our daily lives, we must establish and enforce suitable laws and regulations to safeguard the safety and welfare of all stakeholders. Regrettably, numerous countries lack policies or regulations to address these issues [41]. Consequently, it is essential to conduct research to provide evidence-based support for developing and implementing such policies.

The development, research, adoption, and use of the Metaverse raises various ethical and legal concerns [43]. A crucial ethical aspect is focusing on digital identity, which plays a key role in managing the balance between anonymity and pseudonymity in the Metaverse. Privacy is also a significant consideration, especially for individuals with disabilities dependent on assistive technologies for digital interaction [44]. Additionally, intellectual property poses challenges, notably for content creators and developers relying on copyrighted materials or trademarks in virtual settings [45]. Finally, addressing bias is imperative in the realm of digital accessibility. Another challenge with implementing inclusive design principles in immersive and interactive Metaverse environments is the complexity of creating accessible features that cater to a wide range of disabilities. In designing and developing products, designers and developers need to consider multiple considerations, such as the requirements of individuals with visual and hearing disabilities, mobility disabilities limitations, and cognitive disabilities, as well as the needs of neurodivergent individuals, those with autism, and senior citizens. The extent of accessibility and inclusivity in the Metaverse is greatly influenced by its creators' and developers' biases, viewpoints, and priorities. If these designers possess unconscious biases and stereotypes towards individuals with disabilities, these will be reflected in their product and how people with disabilities are involved in the process [46]. As a result, individuals with disabilities may be forced to accept a suboptimal user experience, while non-disabled individuals are afforded the best possible user experience. The Metaverse is likely to present additional challenges in terms

of privacy and security [47], particularly for individuals with disabilities who depend on assistive devices that might be vulnerable to exploitation. Moreover, the cost of accessing the Metaverse, including the necessity of high-end virtual reality technology, may prove prohibitively expensive for those with limited financial resources [38]. To address these issues, it is crucial to have a deep understanding of the needs and preferences of individuals with disabilities and the technical expertise required to integrate accessible features seamlessly into the Metaverse experience. Additionally, the dynamic and ever-evolving nature of Metaverse environments presents unique challenges in ensuring accessibility [48]. Therefore, it is essential to constantly reassess and update accessibility standards to ensure that individuals with disabilities can fully participate in the Metaverse. Collaboration between accessibility experts, developers, and users with disabilities is critical in this process, as it enables the identification of barriers and the development of innovative solutions. By prioritizing accessibility, the Metaverse can become a more inclusive and empowering space for all users, regardless of their abilities.

## 3. Materials and Methods

### 3.1. Study Design

This present research aims to explore a novel research question that has yet to be addressed in previous research, focusing on the Metaverse and its potential to improve accessibility. To achieve this, a qualitative approach known as the Delphi method was employed as the primary research technique in this study [49]. The Delphi design was chosen due to its capability to promote structured group communication and to reach a consensus on a set of items in a particular area that has not yet been fully explored [50,51]. Delphi's features enabled us to interact with a diverse group of international researchers who were distributed across various geographic locations and to engage with them formally [52].

This study employed a structured Delphi method to systematically gather and refine insights from experts in accessibility, AI, and the Metaverse. Initially, we identified 21 candidates through a meticulous selection process, focusing on their proven expertise and contributions to these fields. Of these, 11 experts agreed to participate. The selection was aimed to ensure a broad representation of views and deep knowledge across the relevant domains. Participants were then provided with a detailed questionnaire designed to elicit their in-depth opinions on several key issues at the intersection of accessibility and the Metaverse. The questionnaire included 13 carefully formulated questions that covered a range of topics from core Metaverse principles to practical challenges in ensuring inclusivity. These were based on their professional experience, academic research, and personal insights into the future of accessible digital environments. After collecting their written responses, we conducted a rigorous qualitative analysis to identify recurring themes, insights, and recommendations. This iterative process of feedback and refinement through the Delphi method allowed us to ensure the credibility and reproducibility of our study by grounding it in the consensus among leading experts. The methodology section of the paper details the qualitative analysis approach, including criteria for expert selection, the nature of the questions posed, and the analytical techniques used to synthesize the data, thereby addressing potential concerns regarding the study's rigor and the foundation for future research.

Furthermore, our research methodology delved into the potential implications of emerging technologies, ethical and legal considerations, and the necessity for collaborative efforts among various stakeholders. This methodological approach was essential in providing a comprehensive understanding of the dynamic interplay between the Metaverse and disability empowerment, guiding the scope and depth of our investigation. The questions posed to the experts are shared in Appendix A. Subsequently, the completed questionnaires were collected, and the facilitator organized and compiled the comments into a consolidated document. This compilation was then shared with each participant, allowing them to provide additional feedback. After each comment session, all questionnaires were returned to the facilitator for evaluation to determine whether further rounds

of feedback were necessary or if the results were ready for publication. This rigorous and iterative Delphi method ensured a thorough exploration and refinement of expert opinions throughout the research process. This approach is effective when exploring lesser-known or under-researched topics, as it allows for incorporating expert perspectives [53]. These opinions then provide a basis for subsequent analysis and screening.

The experts, along with their expertise, are presented in Table 1.

**Table 1.** Eleven experts and their areas of expertise.

|  | Expertise |
|---|---|
| 1 | Metaverse, AI |
| 2 | Education, eLearning, AI |
| 3 | Education, eLearning, AI |
| 4 | Social Sciences, Cognitive Accessibility |
| 5 | AI, Assistive Technology |
| 6 | AI, Education |
| 7 | Digital Accessibility, Human–Computer Interaction |
| 8 | Disability Rights, Assistive Technology |
| 9 | AI |
| 10 | AI, Education, Metaverse, NFC, Blockchain |
| 11 | Metaverse |

### *3.2. Data Analysis*

Thematic analysis was deemed the most appropriate method for analyzing the data in this research, owing to its capability to identify recurring themes and patterns within the responses. The thematic analysis presents a structured and adaptable framework for organizing and categorizing textual data, enabling the recognition of central themes and concepts that emerge naturally from the data [54].

### *3.3. Ethical Considerations*

Participants were fully informed about the study's objectives and procedures prior to completing the questionnaire. All experts were contacted via email and requested to confirm their participation by responding with a return email. This initial communication ensured that all participants were willing volunteers and provided their free and informed consent to participate in the research through electronic means.

## 4. Results and Discussion

In this section, we extensively analyze the Metaverse's role in ensuring digital accessibility, particularly for individuals with disabilities. To begin, we assess the current state of inclusivity in the Metaverse, examining both advancements and challenges associated with integrating assistive technologies. We then delve into the technological solutions and obstacles in creating accessible virtual environments, emphasizing the need for ongoing innovation. Following this, we investigate the ethical and legal implications of digital accessibility in the Metaverse, highlighting the importance of non-discrimination and legal adaptability. Moving forward, we propose a comprehensive framework for an accessible Metaverse, focusing on advanced technologies, user-centric design, and universal access. Finally, we recommend future research directions and policy reforms that can contribute to a more inclusive and equitable virtual world, showcasing the Metaverse's transformative potential while acknowledging its current limitations in inclusivity.

The synthesis of specialized perspectives presents a comprehensive and detailed analysis of the expanding domain of the Metaverse and its ramifications for digital accessibility. This analysis, derived from diverse inquiries, offers a thorough comprehension of the existing status and potential future trajectory of this developing digital landscape, particularly regarding individuals with disabilities. At the core of these discussions is the recognition of the Metaverse as a groundbreaking digital space, marked by its immersive and intercon-

nected virtual environments. The experts concur that the Metaverse harbors the immense potential to transform the experiences of individuals with disabilities. However, they emphasize that actualizing these potentials is contingent upon incorporating accessibility into the very foundations of its development.

The exploration of the key elements and principles of the Metaverse and their complicated connection to digital accessibility reveal crucial insights into the design and implications of this emerging virtual space. Experts and perspectives from diverse domains contribute valuable perspectives, shedding light on the multi-faceted nature of the Metaverse and the imperative role of digital accessibility.

The Metaverse, with its capacity for transformative change, is influenced by essential components and tenets that aim for inclusivity and accessibility. These components and tenets are described in Figure 1.

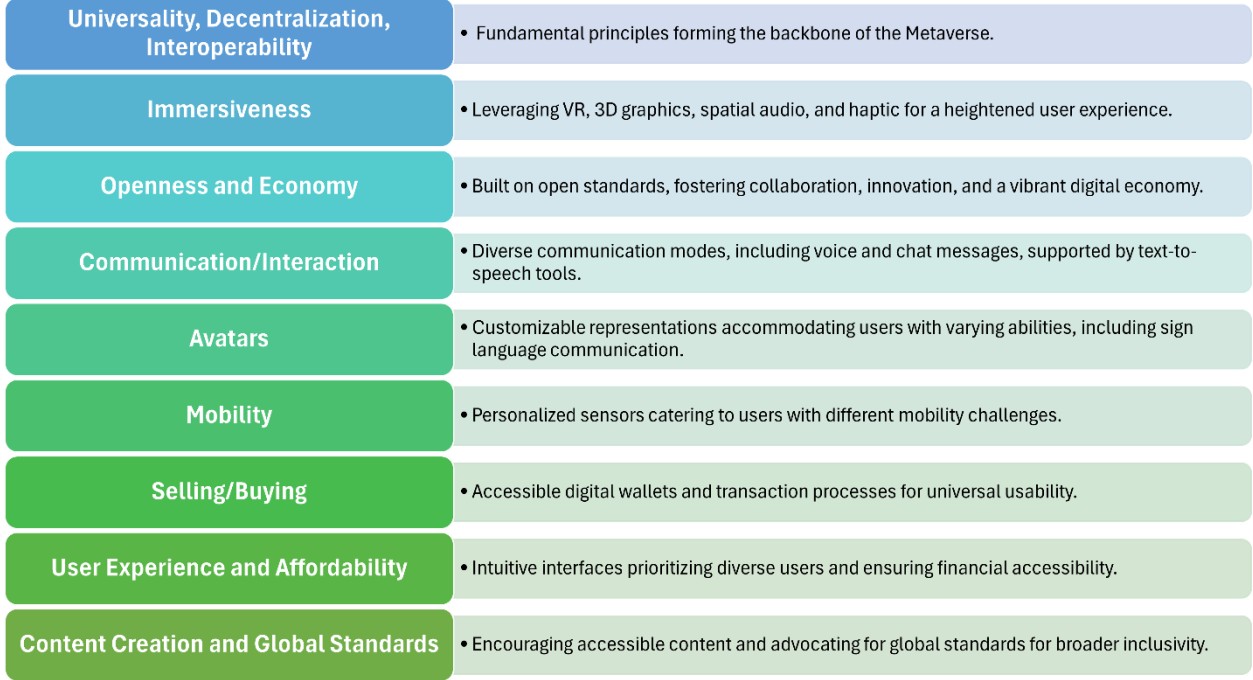

**Figure 1.** Key elements and principles of the accessible Metaverse as per expert opinions.

The mixed state of inclusivity in the present Metaverse is an essential finding from the experts' opinions. While there have been advancements in providing opportunities for individuals with disabilities, the experts also bring attention to notable gaps. One of these is the obstacle of smoothly incorporating assistive technologies and guaranteeing that various virtual settings within the Metaverse can function harmoniously.

The technological and design aspects of the Metaverse are subject to rigorous examination. Experts stress the importance of continuous technological advancement, specifically designed to cater to the distinct requirements of individuals with disabilities. This encompasses tackling the obstacles associated with creating universally accessible virtual spaces and guaranteeing compatibility with a wide range of assistive technologies.

The responses highlight the importance of ethical and legal considerations, with concerns surrounding privacy, autonomy, and the prevention of discrimination being repeatedly expressed. Additionally, there is a demand for legal frameworks to be modified and improved to accommodate the growth of the Metaverse and to guarantee compliance with digital accessibility standards.

In anticipation of future developments, several research avenues have been proposed by experts. These encompass the advancement of assistive technology, the exploration of the user experience for individuals with disabilities in virtual environments, and the

formulation of comprehensive policies and standards that promote accessibility within the Metaverse.

Policy recommendations are also highly informative. The specialists propose amending current digital accessibility regulations, drafting Metaverse-specific rules, and ensuring the participation of individuals with disabilities in decision-making related to the Metaverse's governance and growth.

The synergy of expert opinions portrays a comprehensive landscape of digital accessibility within the Metaverse. Integrating technological innovations, user-centered design, ethical discourse, legal anticipation, and ceaseless research, this narrative presents a holistic view of the Metaverse's potential as a transformative digital platform for individuals with disabilities despite its present constraints in inclusivity.

### 4.1. Inclusivity in the Metaverse

According to the Oxford English Dictionary, inclusivity is "the practice or policy of providing equal access to opportunities and resources for people who might otherwise be excluded or marginalized, such as those having physical or intellectual disabilities or belonging to other minority groups" [55]. The exploration of inclusivity within the Metaverse, as depicted in the insights of various experts, reveals a landscape marked by both promising advancements and notable challenges. The perspectives gathered indicate a shared understanding of the Metaverse's capacity to improve accessibility for persons with disabilities substantially; however, they also emphasize the pressing requirement to address the current deficiencies in its inclusive design.

The Metaverse presents a unique opportunity for individuals with disabilities, as it offers an environment where physical limitations do not inherently pose barriers. Expert 1 highlights this potential, noting that the Metaverse provides tools that support immersive learning experiences and do not discriminate against those with physical disabilities. This perspective is shared by other experts, who believe that the Metaverse has the potential to level the playing field for people with disabilities and provide them with unprecedented opportunities for engagement and interaction.

Achieving an inclusive vision for the Metaverse is not without its challenges. One of the key issues identified by experts is the compatibility of the Metaverse with assistive technologies, which is essential for ensuring accessibility. Expert 4 points out, "Despite the advancements, the Metaverse is yet to be fully accessible, especially for users who rely on assistive technologies". The current state of Metaverse development presents a significant gap in integrating assistive tools and technologies, posing a pressing challenge.

The significance of incorporating inclusive design and development practices within the Metaverse is another commonly mentioned topic among the expert responses. Additionally, the emphasis on involving individuals with disabilities in the development process is highlighted, with Expert 6 stating, "Involving individuals with disabilities in the design process ensures that the Metaverse is built with accessibility at its core". This methodology is considered indispensable not only for constructing inclusive environments but also for guaranteeing that these spaces are instinctive and accommodating for a wide array of users.

The interoperability issue within the Metaverse presents itself as a significant challenge that requires attention. Expert 3 observes, "Interoperability across different virtual environments remains a challenge, impacting the overall accessibility for users with disabilities". This points to the need for a more unified approach in Metaverse development, where seamless movement and interaction across different virtual environments are made possible.

The economic and social consequences of fostering an inclusive Metaverse are crucial aspects of the discourse. On the one hand, an inclusive Metaverse could bring about expanded economic prospects and heightened social involvement for people with disabilities. However, if inclusivity is not prioritized in the development of the Metaverse, there is a potential for social exclusion and economic marginalization. Expert 5 articulates this concern, warning of the potential risks if inclusivity is overlooked.

The perspectives of various specialists indicate that the Metaverse possesses immense potential to revolutionize the lives of individuals with disabilities. However, realizing this potential necessitates addressing the obstacles of interoperability, seamlessly integrating assistive technologies, and maintaining a focus on inclusivity throughout the development process.

### 4.2. Technological Solutions and Challenges

The discussion of technological advancements and difficulties in the context of the Metaverse, as informed by the experts' opinions, highlights a complex interplay between innovation and obstacles. This interplay is crucial in creating a Metaverse that is both accessible and inclusive, particularly for those with disabilities.

The Metaverse has witnessed significant progress in integrating advanced technologies to enhance accessibility. Experts acknowledge these strides, citing the incorporation of assistive tools, such as screen readers and adaptive interfaces, as key advancements. These technologies play a pivotal role in facilitating navigation within the Metaverse, enabling individuals with disabilities to interact with virtual environments more easily. As Expert 1 notes, "The incorporation of screen readers and closed captioning is essential in making the Metaverse more accessible". Furthermore, Expert 10 adds another layer to the discussion, emphasizing a comprehensive approach to accessibility in the Metaverse. The expert shares, "The Metaverse should provide options for text-to-speech and speech-to-text functionalities, haptic feedback, and customizable user interfaces. It should also support screen readers and other assistive technologies".

However, experts also draw attention to the persisting challenges alongside advancements. One significant hurdle is the development of virtual environments that are universally accessible. Despite advancements, a one-size-fits-all solution remains elusive, as Expert 4 points out: "Creating a universally accessible Metaverse is a complex task, owing to the diverse needs of users with disabilities". Expert 11 adds, "Ensuring everyone has access to the necessary hardware and software remains a crucial hurdle. Affordable and adaptable VR equipment, inclusive interface design, and internet connectivity are essential to prevent further marginalization". This complexity is further compounded by the rapid pace of technological advancements in virtual reality and related fields, making it a moving target for developers and designers.

The advancements in improving accessibility in the Metaverse have drawn attention to another significant issue: the need for interoperability and compatibility with a diverse array of assistive technologies and to be fully accessible for persons with disabilities who do not use assistive technologies. This is not only about creating new tools but also ensuring smooth integration with existing technologies. Expert discussions have highlighted the importance of addressing this challenge to realize the potential of the Metaverse fully. Expert 6 underscores the complexity of this issue, stating, "Users who are blind or visually impaired will require screen reading and audio description features in the Metaverse, designed to be a more visually immersive environment than today. This is already a huge challenge to deal with". Expert 5 echoes these concerns by emphasizing the pivotal role of compatibility in the Metaverse. "Ensuring compatibility with various assistive technologies is a key challenge in the Metaverse, requiring ongoing innovation and adaptation". This emphasizes the dynamic nature of the challenge at hand; as technology advances, so must the efforts to ensure compatibility. Continuous innovation highlights the need for a proactive approach to tackle the ever-evolving landscape of assistive technologies.

The recurring theme of the necessity for ongoing research and development to tackle these challenges is evident. As the Metaverse continues to progress, it is essential that the technologies facilitating accessibility within it also evolve. Expert 6 highlights the potential role of generative AI in creating more personalized and adaptive virtual environments, suggesting, "Generative AI could revolutionize the way we approach accessibility in the Metaverse, offering more personalized solutions for users with disabilities".

The Metaverse faces a significant challenge because it incorporates a diverse range of technologies, tools, and environments, which can pose difficulties in achieving the desired

outcome. While there may be some features that are accessible, it is possible that they may not work well for everyone due to potential conflicts in accessibility. In conclusion, the findings from the experts indicate a dynamic landscape where technological advancements for enhancing accessibility in the Metaverse are continually emerging. While substantial progress has been attained, the path toward a completely inclusive Metaverse remains an ongoing endeavor characterized by obstacles that necessitate innovative solutions and an unwavering dedication to perpetual enhancement. The insights shared by these experts emphasize the importance of a coordinated effort in research, development, and design to surmount these challenges and unlock the full potential of the Metaverse for all users.

### 4.3. Ethical and Legal Implications

Examining the ethical and legal ramifications in constructing an inclusive Metaverse, as elucidated by the experts' perspectives, discloses a complicated landscape in which ethical dilemmas and legal obligations intersect substantially with technological progress.

According to the experts, the development of the Metaverse must prioritize ethical considerations, particularly in ensuring accessibility for individuals with disabilities. They emphasize the need for non-discrimination and equal access in this regard. As Expert 1 remarks, "It's essential that the Metaverse is developed with ethical considerations at the forefront, ensuring that no individual is disadvantaged or excluded based on their abilities".

The issue of privacy is of particular concern, particularly for individuals with disabilities who rely on specialized and potentially intrusive assistive technologies. Expert 5 points out the intricacies of privacy in the Metaverse, stating, "Privacy takes on a new dimension in the Metaverse, especially when we consider users who may need assistive technologies that are deeply integrated into their virtual experiences". Moreover, Expert 11 underscores the importance of data privacy and security within the Metaverse, highlighting the highly personal nature of avatars and interactions, which raise concerns about data privacy and the potential for discrimination based on virtual representations. Robust security measures and ethical guidelines are necessary to protect users and prevent misuse.

The legal implications surrounding the Metaverse are equally intricate. Experts suggest that existing digital accessibility regulations, such as the Americans with Disabilities Act (ADA), should be reassessed and potentially expanded to include the distinct characteristics of the Metaverse. Expert 7 underscores this need: "The legal frameworks that govern digital accessibility, such as the ADA, must evolve to stay relevant in the context of the Metaverse's unique challenges and opportunities".

The challenge of establishing universally accepted legal and ethical standards for the Metaverse is a significant concern among experts, given its global nature surpasses traditional legal jurisdictions. This complexity raises concerns about effectively enforcing accessibility standards within the Metaverse. Expert 3 elaborates on this: "Developing and enforcing standards for digital accessibility in the Metaverse is challenging due to its international reach and the rapid pace of technological change".

Experts also discuss the need for governance structures that can oversee the ethical and legal aspects of the Metaverse. They advocate for including stakeholders with disabilities in these governance mechanisms to ensure that their voices are heard and their needs are addressed. As Expert 4 suggests, "Inclusive governance structures are needed to ensure the Metaverse is developed and managed in a way that respects the rights and needs of all users, especially those with disabilities". Expert 8 further reinforces the need for apparent authority in the Metaverse. "In the context of the Metaverse, it's crucial to establish a clear authority responsible for safeguarding the rights and well-being of people with disabilities". This underscores the imperative need for a centralized authority or framework that not only establishes ethical guidelines but also proactively oversees and safeguards the interests of individuals with disabilities.

Additionally, Expert 11 brings attention to another critical aspect, emphasizing the need for careful consideration regarding the ethical implications of embodiment and identity in the Metaverse. "The immersive nature of the Metaverse raises questions about

the blurring lines between real and virtual identities. Careful consideration is needed regarding the implications for mental health, ownership of virtual representations, and potential exploitation".

In conclusion, while expert opinions have highlighted the multi-faceted ethical and legal challenges in crafting an inclusive Metaverse, it is crucial to also acknowledge and address the potential for harassment, which represents a significant barrier to accessibility. This oversight underscores the imperative for a more holistic approach, necessitating ongoing dialogue and collaboration not just among technologists, legal professionals, and policymakers but also with the disability community to mitigate such risks. Achieving ethical and legal integrity in the Metaverse extends beyond technological innovation to encompass a societal commitment to safeguarding inclusivity against all forms of harassment, thereby ensuring a truly accessible digital realm for every user.

*4.4. Framework for an Accessible Metaverse*

The experts have identified a comprehensive framework for an accessible Metaverse. This framework unites a variety of principles and strategies that are designed to cultivate an all-inclusive digital realm. This framework functions as a guiding principle, leading towards a Metaverse in which accessibility is not merely an optional element but a fundamental cornerstone.

- Integrating Advanced Technologies for Accessibility: At the center of this framework is the incorporation of state-of-the-art technologies specifically created to improve accessibility through sophisticated assistive technologies and digital accessibility solutions. This involves utilizing virtual reality (VR) and augmented reality (AR) to surpass conventional accessibility limitations. The aim is to harness these technologies for their immersive properties and capitalize on their potential to create adaptable and responsive environments that cater to the wide-ranging needs of individuals with disabilities. Moreover, cryptocurrency and blockchain can significantly benefit persons with disabilities by enhancing transparency, reducing transaction costs, and providing digital identity solutions. These technologies can facilitate direct money transfers, bypassing intermediaries and reducing graft, which is particularly valuable for individuals relying on remittances or aid. Blockchain's ability to securely and transparently manage transactions and records can also support the creation of more accessible and inclusive financial and social systems for persons with disabilities.

- Emphasizing User-Centric Design: The framework underscores the importance of user-centric design, which revolves around the principle that users, particularly those with disabilities, ought to have a substantial influence on their experiences within the Metaverse. This entails devising environments that are not only accessible but also adaptable to individual preferences and requirements. The focus on personalization and user-friendly interfaces is crucial in this approach, guaranteeing that each user encounters the Metaverse as a realm where their specific needs are addressed through intuitive and uncomplicated means.

- Universal Access as a Foundational Principle: A critical aspect of this framework is the principle of universal access. It emphasizes the importance of designing virtual environments that are inclusive and accessible to individuals of all abilities, promoting a culture of inclusivity that leaves no one behind. This involves a proactive approach to accessibility, incorporating it into the development process from the beginning rather than simply adding it as an afterthought.

- Social and Economic Inclusion: The framework emphasizes the significance of social interaction and economic participation in fostering an inclusive Metaverse. It advocates for establishing a digital economy within the Metaverse that is accessible and inclusive, providing individuals with disabilities with opportunities for economic empowerment and social engagement. This dimension encompasses the broader concept of accessibility, which extends beyond physical and technological barriers to encompass economic and social considerations.

- Global Standards and Governance: Finally, the framework calls for establishing global standards and governance structures that advocate for and enforce accessibility in the Metaverse. Developing universal guidelines that are not limited by geographical boundaries and ensuring that accessibility in the Metaverse is a constant rather than a variable requires a collaborative approach to governance. This approach should involve the participation of various stakeholders, including individuals with disabilities, in decision-making processes. By taking this approach, we can ensure that accessibility in the Metaverse is not just a goal but a reality for all users, regardless of their physical or cognitive abilities.

The framework proposed, depicted in Figure 2, contemplates a Metaverse that is fundamentally grounded in accessibility, encompassing both technological and design elements and social and economic participation. It adopts a comprehensive approach, recognizing the multi-faceted character of accessibility and to tackle it through the fostering of innovation, empathy, inclusivity, and cooperation. This framework serves not only as a roadmap for a more accessible digital realm but also as a call to action for a more inclusive future in the virtual world.

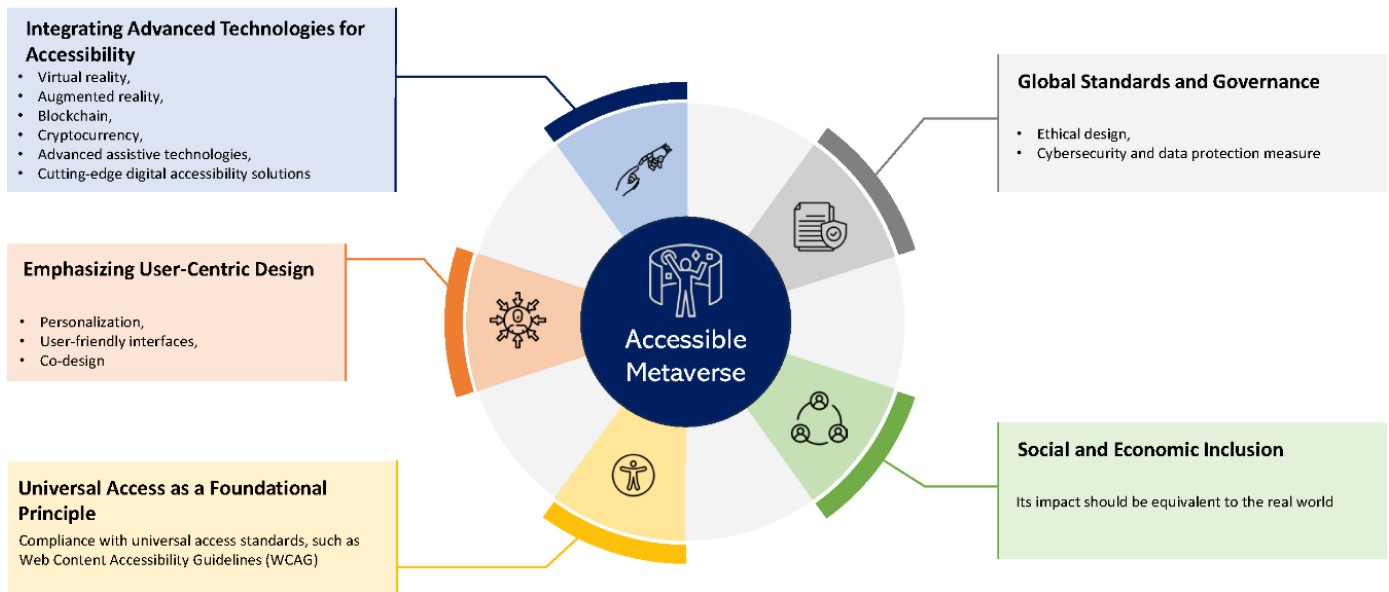

**Figure 2.** Accessible Metaverse framework.

To address the vital need for inclusivity in the Metaverse, our proposed framework delineates a multi-faceted approach, integrating advanced technologies, user-centric design, universal access principles, social and economic inclusion strategies, and global standards for governance. At its core, the framework employs state-of-the-art technologies such as VR and AR to transcend traditional accessibility barriers, offering immersive and adaptable environments tailored to diverse user needs, including those with disabilities. Emphasizing user-centric design, the framework ensures that environments within the Metaverse are developed with direct input from users, especially those with disabilities, allowing for personalization and ease of use. The principle of universal access is foundational, advocating for the inclusion of all individuals from the onset of the development process rather than as an afterthought, thereby embedding accessibility into the fabric of the Metaverse. Social and economic inclusion is promoted by fostering opportunities for economic empowerment and social interaction within a digital economy that is accessible to everyone. Lastly, the establishment of global standards and governance structures aims to advocate and enforce accessibility across all regions, with the involvement of various stakeholders, including those with disabilities, in decision-making processes. This comprehensive framework not only serves as a guideline for creating a more accessible digital realm but also as a call to

action for stakeholders across the spectrum to collaborate towards an inclusive Metaverse. Implementing this framework involves a collaborative effort among developers, policy-makers, accessibility experts, and the community of users with disabilities, ensuring that the development processes of the Metaverse inherently consider and address the diverse needs of all users, thereby making inclusivity a reality in this burgeoning virtual space.

*4.5. Future Research and Policy Recommendations*

The provision of future research and policy recommendations based on expert insights serves as a guide for cultivating an inclusive digital landscape in the Metaverse. These suggestions encompass a broad spectrum of initiatives, from technological advancements to policy reforms.

### 4.5.1. The Support for Continuous Accessible Metaverse Research

In recognizing the importance of continuous advancement in technologies that enhance accessibility in the Metaverse, our research underscores the necessity for specific future investigations. Critical areas requiring further exploration include the development and integration of cutting-edge assistive technologies tailored to meet the varied needs of individuals with disabilities. This entails pioneering the use of artificial intelligence and machine learning to create adaptive and fully inclusive virtual environments. Additionally, there is a pressing need to examine how these technologies can facilitate seamless interaction and navigation within the Metaverse for users with disabilities. To this end, future research should focus on developing practical, user-centered design methodologies that incorporate feedback from individuals with disabilities, ensuring the Metaverse is an accessible, engaging, and empowering space for all users. As Expert 6 suggests, "Investigating the potential of AI and machine learning in creating adaptive virtual environments could be pivotal in making the Metaverse more accessible".

### 4.5.2. Enhancing User Experience for Individuals with Disabilities in the Metaverse

Another significant area for research is the user experience of individuals with disabilities within the Metaverse. Understanding their challenges and preferences is crucial for designing more inclusive virtual spaces. Expert 1 emphasizes this, stating, "Future research should focus on the lived experiences of individuals with disabilities in the Metaverse to tailor the environment to their needs". There is also the risk of sensory overload. Expert 7 highlights this: "The Metaverse may incorporate a range of sensory inputs (visual, auditory, haptic). How can we ensure this doesn't lead to sensory overload, especially for individuals with sensory processing disorders?".

### 4.5.3. Policy Interventions for an Inclusive Metaverse

Policy interventions also play a critical role in shaping an inclusive Metaverse. Experts advocate for developing comprehensive policies that address digital accessibility at a foundational level. This includes revising existing digital accessibility laws and regulations to encompass the unique aspects of the Metaverse. Expert 7 points out, "Current digital accessibility regulations need to be expanded and adapted to ensure they apply to the Metaverse".

### 4.5.4. International Collaboration for Global Accessibility Standards

The establishment of global standards for accessibility in the Metaverse is another area where policy intervention is necessary. Given the Metaverse's global nature, international collaboration in formulating these standards is essential. Expert 3 notes, "Creating universal accessibility standards for the Metaverse requires international cooperation to ensure consistency and efficacy".

4.5.5. Inclusive Governance and Representation in Metaverse Policy

Furthermore, experts call for policies that promote the inclusion of individuals with disabilities in the design and governance of the Metaverse. Ensuring their representation in decision-making processes is key to developing an environment that truly caters to their needs. As Expert 4 articulates, "Policies must be put in place to guarantee the active participation of individuals with disabilities in shaping the Metaverse".

In conclusion, the experts' insights provide a comprehensive outline for future research and policy interventions to cultivate an inclusive digital environment in the Metaverse. These recommendations highlight the importance of continuous technological innovation, the need for empathetic and user-centered research, and the critical role of policy in guiding the development of an accessible and equitable virtual world.

One limitation of our study is the possibility of bias. Despite our efforts to minimize biases, the inherently subjective nature of certain data collected and the biases existing within the experts involved may have impacted the outcome of the study. We prioritized methodological rigor in all aspects of our research to address the limitations of our study. Methodological rigor, through the establishment of clear and transparent criteria for data collection, analysis, and interpretation, is one way to minimize bias in research [56]. In addition, we recommend that future studies in this field consider broadening the pool of experts to include individuals from a variety of diverse backgrounds and perspectives.

## 5. Conclusions

The present study aimed to analyze the current state of inclusivity and accessibility in the Metaverse and identify areas where it could be enhanced. Eleven experts were consulted to provide their perspectives on the Metaverse's current state, its underlying principles, and the challenges and opportunities it presents. The results indicate a mixed level of inclusivity within the Metaverse, with progress made in incorporating assistive technologies and gaps in ensuring interoperability across various virtual environments. A comprehensive theoretical framework is proposed to facilitate further research and inform policy interventions aimed at promoting inclusivity in the Metaverse.

The key findings underscore the pivotal role of integrating individuals with disabilities in the design and development of the Metaverse, as it ensures the incorporation of their unique perspectives into creating an inclusive virtual world. The recommendations emphasize the necessity of ongoing technology innovation to address accessibility challenges while highlighting the need for continuous research and development to keep pace with evolving needs. Accessibility in virtual environments is paramount, and policy plays a pivotal role in ensuring its implementation. A comprehensive policy framework should prioritize inclusivity and foster awareness among developers and users. By investing in training and educational programs, developers are equipped with the necessary knowledge and skills to design accessible virtual worlds. The active involvement of individuals with disabilities, coupled with investments in technology innovation and the implementation of inclusive policies, results in a virtual world that is enriching and empowering for all users.

Continuous research, development, and policymaking are vital to guarantee that the Metaverse remains an inclusive and empowering space. It is essential to stay current with the most recent advancements and accessibility requirements. This entails examining the needs and experiences of individuals with disabilities in virtual settings and developing innovative technologies and tools. Continuous policymaking is necessary to establish guidelines that foster accessibility and inclusivity. The collaboration among technology companies, accessibility experts, and disability advocacy organizations is crucial to addressing specific challenges and developing innovative solutions, paving the way for a more inclusive and accessible Metaverse where all individuals can actively participate, engage, and thrive.

**Author Contributions:** Conceptualization, A.O.; Investigation, K.C.; Methodology, A.O., K.C., A.T. and H.C.; Project administration, A.O.; Supervision, A.O.; Validation, A.E.H., A.T., D.A.-T., F.A. and H.C.; Visualization, A.O. and A.E.H.; Writing—original draft, A.O. and K.C.; Writing—review and editing, A.O., K.C., A.E.H., A.T., C.Y.Z., D.A.-T., F.A., H.C., H.S.A.-K., M.O., M.J., T.A.-H. and Z.A. All authors have read and agreed to the published version of the manuscript.

**Funding:** This research received no external funding.

**Institutional Review Board Statement:** Not applicable.

**Informed Consent Statement:** Informed consent was obtained from all subjects involved in the study.

**Data Availability Statement:** Data are contained within the article.

**Conflicts of Interest:** The authors declare no conflicts of interest.

## Appendix A

The following questions were posed to the experts:

1.  What are the key elements and principles of the Metaverse? How do they relate to digital accessibility?
2.  How do you define digital accessibility within the context of the Metaverse?
3.  What theoretical advances can be made in the field of digital accessibility through the exploration of the Metaverse, and what opportunities and challenges does this present for disability empowerment?
4.  What are the most important theories or models related to digital accessibility in the Metaverse? How can they be integrated into a cohesive framework?
5.  How can we ensure that the Metaverse is designed/co-designed and developed to be inclusive and accessible to all users, regardless of their abilities?
6.  How can the Metaverse be leveraged as a tool for generating insights into the experiences of individuals with disabilities in digital spaces, and what specific use cases illustrate the potential of the Metaverse for advancing our understanding of digital accessibility and disability empowerment?
7.  What impact do emerging technologies such as cryptocurrency and blockchain have on the development of the Metaverse, and how might these advances shape the accessibility and inclusivity of the Metaverse for individuals with disabilities?
8.  What are the existing technological solutions, tools, or best practices for enhancing digital accessibility in the Metaverse?
9.  What are the broader implications of the Metaverse for disability empowerment and digital accessibility, and how might these impact societal attitudes towards disability and accessibility more broadly?
10. To what extent is the Metaverse currently inclusive for people with disabilities, and what specific challenges remain in ensuring that individuals who are visually impaired can fully participate and benefit from this new digital space?
11. How can stakeholders in the Metaverse ecosystem, including developers, policymakers, and disability advocates, collaborate to ensure that the Metaverse creates value and promotes well-being for people with disabilities?
12. Are there any ethical or legal considerations related to digital accessibility in the Metaverse? How can these be addressed within the theoretical framework?
13. What are the most pressing areas for future research in the field of the Metaverse and digital accessibility, and what questions remain unanswered in our understanding of this new frontier?

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
