# Peer review of "Accessible Metaverse: A Theoretical Framework for Accessibility and Inclusion in the Metaverse"

_mti, doi:10.3390/mti8030021_

Round 1

Reviewer 1 Report

Comments and Suggestions for Authors

The manuscript presents a comprehensive framework for understanding and promoting accessibility in Metaverse. Also, the manuscript explore the potential of the Metaverse to improve accessibility for people with disabilities. Overall, the study provides a starting point for understanding accessibility in the Metaverse. However, there are several concerns regarding user study.

The study does not include any data from people with disabilities about their experiences in the Metaverse. Also, the study does not address the potential for harassment in the Metaverse, which could be a barrier to accessibility for some users. This is an important issue to consider, as it could have a significant impact on the inclusivity of the Metaverse. Please discuss this limitation. 

In section 4.4 Framework for an accessible Metaverse describe how blockchain and cryptocurrencies presented in Fig 2 can improve Metaverse accessibility.

Figure 2 is not referenced in the text.

Regards

Author Response

Comments – Reviewer 1

Action

The manuscript presents a comprehensive framework for understanding and promoting accessibility in Metaverse. Also, the manuscript explore the potential of the Metaverse to improve accessibility for people with disabilities. Overall, the study provides a starting point for understanding accessibility in the Metaverse. However, there are several concerns regarding user study.

No action is required.

The study does not include any data from people with disabilities about their experiences in the Metaverse.

The paper is based on Expert opinions as a first step. We’re planning to extend the research to persons with disabilities through a public survey. To answer to the comment, we updated the section 1 Introduction as follow:

Considering the background presented, this paper primarily aims to investigate the current state of accessibility and inclusivity in the Metaverse through expert opinions and analysis of existing platforms and technologies. While acknowledging the importance of diverse user experiences, our study, at this stage, does not include direct data from people with disabilities regarding their experiences in the Metaverse. Recognizing this as a significant limitation, we intend to address this gap in future research phases, where we will specifically focus on collecting and analyzing firsthand experiences from individuals with disabilities to further our understanding and enhance the inclusivity of the Metaverse.

Also, the study does not address the potential for harassment in the Metaverse, which could be a barrier to accessibility for some users. This is an important issue to consider, as it could have a significant impact on the inclusivity of the Metaverse. Please discuss this limitation.

The harassment issue has been included in 4.3 as follow:

In conclusion, while expert opinions have highlighted the multifaceted ethical and legal challenges in crafting an inclusive Metaverse, it's crucial to also acknowledge and address the potential for harassment, which represents a significant barrier to accessibility. This oversight underscores the imperative for a more holistic approach, necessitating ongoing dialogue and collaboration not just among technologists, legal professionals, and policymakers, but also with the disability community to mitigate such risks. Achieving ethical and legal integrity in the Metaverse extends beyond technological innovation to encompass a societal commitment to safeguarding inclusivity against all forms of harassment, thereby ensuring a truly accessible digital realm for every user

In section 4.4 Framework for an accessible Metaverse describe how blockchain and cryptocurrencies presented in Fig 2 can improve Metaverse accessibility.

The blockchain and Cryptocurrency have been explained in the section 4.4 as follow:

Integrating Advanced Technologies for Accessibility:  At the center of this framework is the incorporation of state-of-the-art technologies specifically created to improve accessibility through sophisticated assistive technologies and digital accessibility solutions. This involves utilizing Virtual Reality (VR) and Augmented Reality (AR) to surpass conventional accessibility limitations. The aim is to harness these technologies for their immersive properties and capitalize on their potential to create adaptable and responsive environments that cater to the wide-ranging needs of individuals with disabilities. Moreover, cryptocurrency and blockchain can significantly benefit persons with disabilities by enhancing transparency, re-ducing transaction costs, and providing digital identity solutions. These technolo-gies can facilitate direct money transfers, bypassing intermediaries and reducing graft, which is particularly valuable for individuals relying on remittances or aid. Blockchain's ability to securely and transparently manage transactions and records can also support the creation of more accessible and inclusive financial and social systems for persons with disabilities.

Figure 2 is not referenced in the text.

Figure number was wrong in section 4.4. It is fixed as follow:

The framework proposed, depicted in Figure  2, contemplates a Metaverse that is fundamentally grounded in accessibility, encompassing both technological and de-sign elements and social and economic participation. It adopts a comprehensive approach, recognizing the multifaceted character of accessibility and to tackle it through the fostering of innovation, empathy, inclusivity, and cooperation. This framework serves not only as a roadmap for a more accessible digital realm but also as a call to action for a more inclusive future in the virtual world.

Reviewer 2 Report

Comments and Suggestions for Authors

The article entitled "Accessible Metaverse: A Theoretical Framework for Accessibility and Inclusion in the Metaverse" is relevant to the mission of the journal, as it contributes to increasing the field of knowledge of the so-called Metaverse and its potential for improving accessibility. I really enjoyed reading the paper, as its story line contributes to the reader's motivation.

The abstract of the article is adequate and briefly explains the content of the study and contains all the sections of a scientific article.

The keywords are also adequate.

The title of the article "Accessible Metaverse: A Theoretical Framework for Accessibility and Inclusion in the Metaverse" corresponds to the content of the paper, which analyses the current state of inclusion and accessibility in the Metaverse, and identifies possible areas for improvement. 

The document is well structured, facilitating the understanding of the study. The theoretical underpinning is based on the background of other recent studies carried out along the same lines in different contexts. Likewise, the authors explain why their study is important and justify it on the basis of the future construction of more inclusive metaverses.

The research questions are well formulated and correspond to the objective of the study, which is to investigate the current state of accessibility and inclusion in the Metaverse and to identify areas for possible improvement.

Method: The methodological design is based on a qualitative approach known as the Delphi method as the main research technique. The rationale for the design is also presented.

Ethical considerations are presented.

The research phases are presented in a clear and structured way. 

The figures and tables used contribute to the reader's understanding.

Results: this evaluator considers that the results shown in terms of the study problem are adequately presented, as well as the discussion of these results with previous studies. 

Limitations and future lines of study are included in the study.

In short, I consider that this is a good work that will undoubtedly contribute to an improvement in the design and construction of more inclusive Metaverses.

Author Response

Comments – Reviewer 2

Action

The article entitled "Accessible Metaverse: A Theoretical Framework for Accessibility and Inclusion in the Metaverse" is relevant to the mission of the journal, as it contributes to increasing the field of knowledge of the so-called Metaverse and its potential for improving accessibility. I really enjoyed reading the paper, as its story line contributes to the reader's motivation.

No action is required.

The abstract of the article is adequate and briefly explains the content of the study and contains all the sections of a scientific article.

No action is required.

The keywords are also adequate.

No action is required.

The title of the article "Accessible Metaverse: A Theoretical Framework for Accessibility and Inclusion in the Metaverse" corresponds to the content of the paper, which analyses the current state of inclusion and accessibility in the Metaverse, and identifies possible areas for improvement.

No action is required.

The document is well structured, facilitating the understanding of the study. The theoretical underpinning is based on the background of other recent studies carried out along the same lines in different contexts. Likewise, the authors explain why their study is important and justify it on the basis of the future construction of more inclusive metaverses.

No action is required.

The research questions are well formulated and correspond to the objective of the study, which is to investigate the current state of accessibility and inclusion in the Metaverse and to identify areas for possible improvement.

No action is required.

Method: The methodological design is based on a qualitative approach known as the Delphi method as the main research technique. The rationale for the design is also presented.

No action is required.

Ethical considerations are presented.

No action is required.

The research phases are presented in a clear and structured way.

No action is required.

The figures and tables used contribute to the reader's understanding.

No action is required.

Results: this evaluator considers that the results shown in terms of the study problem are adequately presented, as well as the discussion of these results with previous studies.

No action is required.

Limitations and future lines of study are included in the study.

No action is required.

In short, I consider that this is a good work that will undoubtedly contribute to an improvement in the design and construction of more inclusive Metaverses.

Thank you.

Reviewer 3 Report

Comments and Suggestions for Authors

The article discusses the potential of the Metaverse to enhance digital accessibility for individuals with disabilities, incorporating insights from experts across various fields. The abstract indicates a qualitative analysis of expert responses on digital accessibility and Metaverse development, aiming to illuminate the current state of inclusivity within the Metaverse. The study highlights both progress and existing gaps in accessibility, particularly in the integration of assistive technologies and interoperability across virtual environments. A comprehensive framework is proposed, focusing on technological innovation, user-centric design, universal access, and adherence to global standards, aiming to support future research and policy-making for an inclusive digital environment in the Metaverse. The paper contributes to the discourse on Metaverse and digital accessibility, advocating for a holistic approach that includes technological innovation, user-centered design, ethical considerations, legal compliance, and ongoing research. The paper illustrates the issues well and gives proper emphasis on related issues. The “flow” of this paper is good. The paper is well structured and the problem presented is relevant and current. However, it should provide specific examples or case studies within the Metaverse that illustrate both the advancements and the gaps in accessibility to give the reader a clearer understanding of the current landscape. It should also elaborate on the qualitative analysis methodology, including how experts were selected, the nature of the questions posed, and how responses were analyzed to ensure the study's credibility and reproducibility. The paper should detail the proposed framework's components more clearly, explaining how each element contributes to fostering inclusivity and how it can be implemented within the Metaverse's development processes. It should also expand on the identified ethical and legal considerations by providing examples of potential challenges and offering suggestions for addressing these within the Metaverse to ensure privacy, non-discrimination, and compliance with evolving legal frameworks. Regarding future research directions, the paper may offer more concrete directions for future research, specifying areas within Metaverse development that require further investigation to enhance accessibility. Concerning the interdisciplinary approach, it should encourage collaboration across disciplines, noting how insights from disability studies, technology development, legal studies, and ethical philosophy can converge to enrich the Metaverse's inclusivity. And also, stress the importance of involving individuals with disabilities in the design and testing phases of Metaverse environments to ensure that accessibility features meet actual needs and preferences.

Author Response

Comments – Reviewer 3

Action

The article discusses the potential of the Metaverse to enhance digital accessibility for individuals with disabilities, incorporating insights from experts across various fields.

No action is required.

The abstract indicates a qualitative analysis of expert responses on digital accessibility and Metaverse development, aiming to illuminate the current state of inclusivity within the Metaverse.

No action is required.

The study highlights both progress and existing gaps in accessibility, particularly in the integration of assistive technologies and interoperability across virtual environments. A comprehensive framework is proposed, focusing on technological innovation, user-centric design, universal access, and adherence to global standards, aiming to support future research and policy-making for an inclusive digital environment in the Metaverse.

No action is required.

The paper contributes to the discourse on Metaverse and digital accessibility, advocating for a holistic approach that includes technological innovation, user-centered design, ethical considerations, legal compliance, and ongoing research. The paper illustrates the issues well and gives proper emphasis on related issues.

No action is required.

The “flow” of this paper is good. The paper is well structured and the problem presented is relevant and current.

No action is required.

However, it should provide specific examples or case studies within the Metaverse that illustrate both the advancements and the gaps in accessibility to give the reader a clearer understanding of the current landscape.

Starting from line 215 till 287, we addressed the gap. As there is no clear examples or case studies to refer, we updated the section as follow:

In examining digital accessibility for individuals with disabilities, the prevalent challenges identified were not only technical issues, security and privacy concerns, and operational hurdles but also a notable absence of case studies or practical examples addressing the gap in creating an accessible Metaverse. This omission underscores the need for real-world applications and solutions to better understand and tackle these barriers.

It should also elaborate on the qualitative analysis methodology, including how experts were selected, the nature of the questions posed, and how responses were analyzed to ensure the study's credibility and reproducibility.

The section 3.1. has been fixed as follow to answer the comment. This revision aims to provide a clear and comprehensive description of the Delphi method used, highlighting the systematic approach to expert selection, the detailed nature of the inquiry, and the methodical analysis of responses. It addresses the reviewer's concerns by elaborating on the qualitative methodology, ensuring the study's credibility and the potential for reproduction of its findings:

This study employed a structured Delphi method to systematically gather and re-fine insights from experts in accessibility, AI, and the Metaverse. Initially, we identified 21 candidates through a meticulous selection process, focusing on their proven expertise and contributions to these fields. Of these, 11 experts agreed to participate. The selection was aimed to ensure a broad representation of views and deep knowledge across the relevant domains. Participants were then provided with a detailed questionnaire designed to elicit their in-depth opinions on several key issues at the intersection of accessibility and the Metaverse. The questionnaire included 13 carefully formulated questions that covered a range of topics from core Metaverse principles to practical challenges in ensuring inclusivity. These were based on their professional experience, academic research, and personal insights into the future of accessible digital environments. After collecting their written responses, we conducted a rigorous qualitative analysis to identify recurring themes, insights, and recommendations. This iterative process of feedback and refinement through the Delphi method allowed us to ensure the credibility and reproducibility of our study by grounding it in the consensus among leading experts. The methodology section of the paper details the qualitative analysis approach, including criteria for expert selection, the nature of the questions posed, and the analytical techniques used to synthesize the data, thereby addressing potential concerns regarding the study's rigor and the foundation for future research.

The paper should detail the proposed framework's components more clearly, explaining how each element contributes to fostering inclusivity and how it can be implemented within the Metaverse's development processes.

We extend the section 4.4 with the following justification to detail clearly the framework components:

To address the vital need for inclusivity in the Metaverse, our proposed framework delineates a multi-faceted approach, integrating advanced technologies, user-centric design, universal access principles, social and economic inclusion strategies, and global standards for governance. At its core, the framework employs state-of-the-art technologies such as VR and AR to transcend traditional accessibility barriers, offering immersive and adaptable environments tailored to diverse user needs, including those with disabilities. Emphasizing user-centric design, the framework ensures that environments within the Metaverse are developed with direct input from users, especially those with disabilities, allowing for personalization and ease of use. The principle of universal access is foundational, advocating for the inclusion of all individuals from the onset of the development process, rather than as an afterthought, thereby embedding accessibility into the fabric of the Metaverse. Social and economic inclusion is promoted by fostering opportunities for economic empowerment and social interaction within a digital economy that is accessible to everyone. Lastly, the establishment of global standards and governance structures aims to advocate and enforce accessibility across all regions, with the involvement of various stakeholders, including those with disabilities, in decision-making processes. This comprehensive framework not only serves as a guideline for creating a more accessible digital realm but also as a call to action for stakeholders across the spectrum to collaborate towards an inclusive Metaverse. Implementing this framework involves a collaborative effort among developers, policymakers, accessibility experts, and the community of users with disabilities, ensuring that the development processes of the Metaverse inherently consider and address the diverse needs of all users, thereby making inclusivity a reality in this burgeoning virtual space.

It should also expand on the identified ethical and legal considerations by providing examples of potential challenges and offering suggestions for addressing these within the Metaverse to ensure privacy, non-discrimination, and compliance with evolving legal frameworks.

Section 4.3 has been updated to address more ethical and legal considerations.

Regarding future research directions, the paper may offer more concrete directions for future research, specifying areas within Metaverse development that require further investigation to enhance accessibility.

The section 4.5.1 addressed this point. We extended it by specifying more concrete directions as follow:

In recognizing the importance of continuous advancement in technologies that enhance accessibility in the Metaverse, our research underscores the necessity for specific future investigations. Critical areas requiring further exploration include the development and integration of cutting-edge assistive technologies tailored to meet the varied needs of individuals with disabilities. This entails pioneering the use of artificial intelligence and machine learning to create adaptive and fully inclusive virtual environments. Additionally, there's a pressing need to examine how these technologies can facilitate seamless interaction and navigation within the Metaverse for users with disabilities. To this end, future research should focus on developing practical, user-centered design methodologies that incorporate feedback from individuals with disabilities, ensuring the Metaverse is an accessible, engaging, and empowering space for all users.

Concerning the interdisciplinary approach, it should encourage collaboration across disciplines, noting how insights from disability studies, technology development, legal studies, and ethical philosophy can converge to enrich the Metaverse's inclusivity. And also, stress the importance of involving individuals with disabilities in the design and testing phases of Metaverse environments to ensure that accessibility features meet actual needs and preferences.

The paper is based on Expert opinions as a first step. We’re planning to extend the research to persons with disabilities through a public survey. To answer to the comment, we updated the section 1 Introduction as follow:

Considering the background presented, this paper primarily aims to investigate the current state of accessibility and inclusivity in the Metaverse through expert opinions and analysis of existing platforms and technologies. While acknowledging the importance of diverse user experiences, our study, at this stage, does not include direct data from people with disabilities regarding their experiences in the Metaverse. Recognizing this as a significant limitation, we intend to address this gap in future research phases, where we will specifically focus on collecting and analyzing firsthand experiences from individuals with disabilities to further our understanding and enhance the inclusivity of the Metaverse.

Round 2

Reviewer 1 Report

Comments and Suggestions for Authors

Authors addressed all my comments and suggestions from previous review round. I have no further comments to add.

Regards

Comments on the Quality of English Language

There are no apparent English problems.